# Deep Assessment Methodology Using Fractional Calculus on Mathematical Modeling and Prediction of Gross Domestic Product per Capita of Countries

**Ertuğrul Karaçuha**[ID]**, Vasil Tabatadze**[ID]**, Kamil Karaçuha \***[ID]**, Nisa Özge Önal**[ID] **and Esra Ergün**

Informatics Institute, Istanbul Technical University, 34467 Istanbul, Turkey; karacuhae@itu.edu.tr (E.K.);
vasilitabatadze@gmail.com (V.T.); onal16@itu.edu.tr (N.Ö.Ö.); ergunesr@itu.edu.tr (E.E.)
**\*** Correspondence: karacuha17@itu.edu.tr

**Abstract:** In this study, a new approach for time series modeling and prediction, "deep assessment methodology," is proposed and the performance is reported on modeling and prediction for upcoming years of Gross Domestic Product (GDP) per capita. The proposed methodology expresses a function with the finite summation of its previous values and derivatives combining fractional calculus and the Least Square Method to find unknown coefficients. The dataset of GDP per capita used in this study includes nine countries (Brazil, China, India, Italy, Japan, the UK, the USA, Spain and Turkey) and the European Union. The modeling performance of the proposed model is compared with the Polynomial model and the Fractional model and prediction performance is compared to a special type of neural network, Long Short-Term Memory (LSTM), that used for time series. Results show that using Deep Assessment Methodology yields promising modeling and prediction results for GDP per capita. The proposed method is outperforming Polynomial model and Fractional model by 1.538% and by 1.899% average error rates, respectively. We also show that Deep Assessment Method (DAM) is superior to plain LSTM on prediction for upcoming GDP per capita values by 1.21% average error.

**Keywords:** deep assessment; fractional calculus; least squares; modeling; GDP per capita; prediction; LSTM

---

## 1. Introduction

In the last quarter of the century, the data exchange with not only person to person but also, machine to machine has increased tremendously. Developments in technology and informatics in parallel with the development of data science lead the companies, institutions, universities and especially, the countries to give priority to evaluating produced data and predicting what can be forthcoming. The modeling of all technical, economic, social events and data has been the interest of scientists for many years [1–4]. Many authors have been investigating the modeling and predicting events, options, choices and data. Especially, there is a huge research interest in finding any relation between telecommunication, economic growth and financial development [5–12]. One of the approaches to model a physical phenomenon or a mathematical study is to model the dependent variable satisfying differential equation with respect to the independent variable. However, the differential equations with an integer-order proposed for mathematical economics or data modeling cannot describe processes with memory and non-locality because the integer-order derivatives have the property of the locality. On the other hand, the fractional-order differential equation is a branch of mathematics that focuses on fractional-order differential and integral operators and can be used to address the limitations of integer order differential models. Using the fractional calculus or converting the integer-order differential equation into the non-integer order differential equations lead to a very essential advantage which is

memory property of the fractional-order derivative. This is very crucial for models related to economics which in general, deal with the past and the effect of the past and now on future [12,13]. The memory capability of the fractional differential approach is the foundation of our motivation.

Fractional calculus (FC) as a question to Wilhelm Leibniz (1646–1716) first arose in 1695 from French Mathematician Marquis de L'Hopital (1661–1704) [11]. The main question of interest was what if the order of derivative were a real number instead of an integer. After that, the FC idea has been developed by many mathematicians and researchers throughout the eighteenth and nineteenth centuries. Now, there exist several definitions of the fractional-order derivative, including Grünwald-Letnikov, Riemann-Liouville, Weyl, Riesz and the Caputo representation. The fractional approach is used in many studies because the fractional derivative represents the intermediate states between two known states. For example, zero order-derivative of the function means the function itself while the first-order derivative represents the first derivative of the function. Between these known states, there are infinite intermediate states [11]. The use of semi-derivatives and integrals in the mass and heat transfer become an important instant in the field of fractional calculus due to employing the mathematical definitions into physical phenomena [12,13]. In the last decade, using fractional operators which explain the events, situations or modes between two different stages or the phenomena with memory provide more accurate models in many branches of science and engineering including chemistry, biology, biomedical devices, nanotechnology, diffusion, diffraction and economics [12–31]. In References [25–31], the modeling and comparison of the countries and trends in the sense of economics and its parameters are implemented. In References [25,26], economic processes with memory are discussed and modeling is obtained by using the fractional calculus. The studies with similar purposes as we aim such as modeling or prediction exist. In these studies, the fractional calculus is employed to model the given dataset and to predict for the forthcoming. In Reference [28], the orthogonal distance fitting method is used. The study is trying to minimize the sum of the orthogonal distance of data points in order to obtain an optimized continuous curve representing the data points. In Reference [32], the one-parameter fractional linear prediction is studied using the memory of two, three or four samples, without increasing the number of predictor coefficients defined in the study. In Reference [33], the generalized formulation of the optimal low-order linear prediction by using the fractional calculus approach is developed with restricted memory. All these studies focus on modeling or prediction for a phenomenon with fractional calculus. Also, in our previous studies, methods based on FC that works for modeling were introduced. In these studies, the children's physical growth, subscriber's numbers of operators, GDP per capita were modeled and compared with other modeling approaches such as Fractional Model-1 and Polynomial Models [34–36]. According to the results, proposed fractional models had better results compared to the results obtained from Linear and Polynomial Models [34–36]. Our previous works do not take into account the previous values of the dataset for any time instant. Their purpose is to model the dataset with minimum error and faster way compared with classical methods such as Polynomial and Linear Regression.

In this study, we extend our prior works by predicting the next incoming values as well as modeling the data itself. We introduce a new mathematical model, namely "Deep Assessment," based on the fractional differential equation for modeling and prediction by using the properties of fractional calculus. Different to the literature and our early studies mentioned above, this model can be used for prediction as well as modeling. The proposed approach is built on the fractional-order differential equation and corresponding Laplace transform properties are utilized. Here, the modeling is implemented with mathematical tools similar to those developed in the previous study [4] with a different approach in which the finite numbers of previous values and the derivatives are taken into account. Then, the prediction is obtained by assuming a value in a specific time can be expressed as the summation of the previous values weighted by unknown coefficients and the function to be modeled is continuous and differentiable. In this way, the proposed method takes previous values and variation rates between different time samples (derivative) of the dataset into account while modeling

the data itself and predicting upcoming values. Combining the previous values with the variations weighted by the unknown coefficients lead to calling the method "deep assessment."

In this study, we assessed the proposed method by the modeling, testing and predicting GDP per capita of the following countries and the European Union: Brazil, China, European Union, India, Italy, Japan, the UK, the USA, Spain and Turkey. GDP per capita is a measure of a country's total economic output divided by the number of the population of the country. In general, it is a reasonable and good measurement of a country's living quality and standard [37]. Therefore, the modeling of GDP per capita is crucial and predicting GDP per capita is very essential not only for researchers but also for companies, investors, manufacturers and institutions. To assess the performance of Deep Assessment in modeling, we compare the proposed model with Polynomial Regression and Fractional Model-1 [34]. Besides, in the same way, for the prediction, we compared the model with Long-Short Term Memory (LSTM), a special type of neural networks used in time series problems.

The structure of the study is the following. Section 2 explains the formulation of the problem. After that, Section 3, namely *Our Approach*, is devoted to explaining how to obtain modeling, simulation, testing and prediction. Then, in Section 4, the results are presented. Lastly, Section 5 highlights the conclusion of the study.

## 2. Formulation of the Problem

In this section, the mathematical foundation of the proposed method is given. Before going into the mathematical manipulations, it is better to explain the approach and the main steps for the formulation. The study aims to model and then, to predict GDP per capita data at any time $t$ by using the previous GDP per capita values of the countries. Here, we assume that countries' historical data and the change of these data over time create an eco-genetics for the forthcoming. In other words, mathematically, GDP per capita at a time $t$ is assumed to be the summation of both its previous values and the changes in time with unknown constant coefficients. In the second stage, we express a function for the GDP per capita as a series expansion by using Taylor expansion of a continuous and bounded function. Then, the differential equation obtained from this series expansion is defined. After that, the unknown constant coefficients are found by the least-squares method. The method aims to minimize the error between the proposed GDP per capita function and the dataset.

First, it is a reasonable idea to approximate a function $g(x)$ as the finite summation of the previous values of the same function weighted with unknown coefficients $\alpha_k$ and the summation of the derivatives of the previous values of the same function weighted with unknown coefficients $\beta_k$ because, intuitively, the recent value of data, in general, is related to and correlated with its previous values and the change rates. The purpose is to find the upcoming values of any dataset with a minimum error by employing the previously inherited features of the dataset. As a starting point, an arbitrary function is assumed to be approximately the finite summation of the previous values and the change rates weighted with some constant coefficients. To use the heritability of fractional calculus, this presupposition for modeling of the function itself and predicting future values is done [6,28,34].

$$g(x) \cong \sum_{k=1}^{l} \alpha_k g(x-k) + \sum_{k=1}^{l} \beta_k g'(x-k). \tag{1}$$

Here, $g'$ is the first derivative of $g(x-k)$ with respect to $x$. After assuming Equation (1), the function $g(x)$ can be expanded as the summation of polynomials with unknown constant coefficients, $a_n$ as given in Equation (2). Here, $g(x)$ is assumed to be a continuous and differentiable function.

$$g(x) = \sum_{n=0}^{\infty} a_n x^n. \tag{2}$$

Then, $g(x - k)$ becomes as Equation (3)

$$g(x - k) = \sum_{n=0}^{\infty} a_n (x - k)^n \tag{3}$$

The final form of $g(x)$ is given as Equation (4).

$$g(x) \cong \sum_{k=1}^{l} \alpha_k \sum_{n=0}^{\infty} a_n (x - k)^n + \sum_{k=1}^{l} \beta_k \sum_{n=0}^{\infty} a_n n (x - k)^{n-1}. \tag{4}$$

After combining $\alpha_k a_n$ as $a_{kn}$, $\beta_k a_n$ as $b_{kn}$ and approximating Equation (4), Equation (5) is obtained. Here, truncation of $\infty$ to $M$ is performed. After truncation, the first derivative of $g(x)$ is taken and given in Equation (6).

$$g(x) \cong \sum_{k=1}^{l} \sum_{n=0}^{M} a_{kn} (x - k)^n + \sum_{k=1}^{l} \sum_{n=0}^{M} b_{kn} n (x - k)^{n-1} \tag{5}$$

$$\frac{dg(x)}{dx} \cong \sum_{k=1}^{l} \sum_{n=1}^{M} a_{kn} n (x - k)^{n-1} + \sum_{k=1}^{l} \sum_{n=1}^{M} b_{kn} n (n - 1)(x - k)^{n-2}. \tag{6}$$

The expression given in Equation (7) is the definition of Caputo's fractional derivative [11]. Throughout the study, Caputo's description of the fractional derivative is employed.

$$\mathfrak{D}_x^{\gamma} g(x) = \frac{d^{\gamma} g(x)}{dx^{\gamma}} = \frac{1}{\Gamma(n - \gamma)} \int_0^x \frac{g^{(n)}(k) dk}{(x - k)^{\gamma - n + 1}}, \quad (n - 1 < \gamma < n). \tag{7}$$

In Equation (7), $\Gamma(1 - \gamma)$ is the Gamma function, the fractional derivative is taken with respect to $x$ in the order of $\gamma$ and $g^{(n)}$ corresponds to the $n^{th}$ derivative again, with respect to $x$. In our study, $n = 1$ is assumed and the fractional-order spans between 0 and 1. Here, two expansions are done to express $g(x)$, approximately. The first one is to express the function as the finite summation of the previous values of the function. Second, expressing the function $g(x)$ as the summation of polynomials known as Taylor Expansion assuming that $g(x)$ is a continuous and differentiable function.

Finally, the mathematical background is enough to go further in the proposed methodology. As a summary, above, we mentioned three important tools. First, a function is expressed as the summation of its previous samples. Second, Taylor expansion for a continuous and differentiable function is defined. After that, the Caputo definition of the fractional derivative is given. Now, it is time to express Deep Assessment Methodology by using fractional calculus for the modeling and prediction. Apart from above, there is an assumption that the fractional derivative $f(x)$ in the order of $\gamma$ is equal to Equation (8). After this assumption, it is required to find unknown $f(x)$ which satisfies the fractional differential equation below and models the discrete dataset.

$$\frac{d^{\gamma} f(x)}{dx^{\gamma}} \cong \sum_{k=1}^{l} \sum_{n=1}^{\infty} a_{kn} n (x - k)^{n-1} + \sum_{k=1}^{l} \sum_{n=1}^{\infty} b_{kn} n (n - 1)(x - k)^{n-2}, \tag{8}$$

where, $f(x)$ stands for the GDP per capita of the countries and $x$ corresponds to the time.

Note that, in (6), allowing the order of the derivation in the left-hand side of Equation (6) to be non-integer gives a more general model [28]. This generalization is employed in Deep Assessment Methodology for $f(x)$ which stands for the GDP per capita.

Here, the motivation is to find $a_{kn}$ and $b_{kn}$ given in Equation (8). To find the unknowns, the differential equation needs to be solved. The strategy is as follows—first, it is required to take the Laplace transform which leads to having an algebraic equation instead of a differential equation. In other words, the Laplace transform is taken for Equation (8) to reduce the differential equation

to algebraic equation, then, by using inverse Laplace transform properties, the final form of $f(x)$ is obtained as Equation (9) [11].

$$f(x,\gamma) \cong f(0) + \sum_{k=1}^{l} \sum_{n=1}^{\infty} a_{kn} C_{kn}(x,\gamma) + \sum_{k=1}^{l} \sum_{n=1}^{\infty} b_{kn} D_{kn}(x,\gamma),$$
where,
$$C_{kn}(x,\gamma) \triangleq \frac{\Gamma(n+1)}{\Gamma(n+\gamma)}(x-k)^{n+\gamma-1}$$
$$D_{kn}(x,\gamma) \triangleq \frac{\Gamma(n+1)}{\Gamma(n+\gamma-1)}(x-k)^{n+\gamma-2}.$$
(9)

To obtain the numerical calculation, the infinite summation of polynomials is approximated as a finite summation given in Equation (10).

$$f(x,\gamma) \cong f(0) + \sum_{k=1}^{l} \sum_{n=1}^{M} a_{kn} C_{kn}(x,\gamma) + \sum_{k=1}^{l} \sum_{n=1}^{M} b_{kn} D_{kn}(x,\gamma).$$
(10)

Here, $f(0)$, $a_{kn}$ and $b_{kn}$ are unknown coefficients that need to be determined. Note that, below, properties of the Laplace transform ($\mathcal{L}$) are given to find Equations (9) and (10) [11].

$$\mathcal{L}\left[(x-k)^{n-1}\right] = \frac{\Gamma(n)}{s^n}e^{-ks} \text{ and } \mathcal{L}\left[\frac{d^\gamma f(x)}{dx^\gamma}\right] = s^\gamma F(s) - s^{\gamma-1}f(0) \text{ for } 0 < \gamma < 1.$$

where, $\mathcal{L}$ stands for the Laplace transform and $\mathcal{L}[f(x)] = F(s)$.

For the numerical calculation, the infinite summation is converted into a finite summation, as given in Equation (10).

## 3. Our Approach

### 3.1. Modeling with Deep Assessment

In this part, the methodology for the modeling of the problem is given in detail. To predict the upcoming years, the problem has four regions as given in Figure 1. Dataset spans in Region 1, 2 and 3. Note that, there is no data for Region 4 where the prediction is aimed. Region 1 is called "before modeling region" which consists of historical data. Each of the coefficients $(x-k)^{n+\gamma-1}$ and derivative coming from previous values of GDP per capita for different values of $k$ and multiplication by different weights as given in Equation (10) will add the contribution to the recent data. For modeling, the historical data is employed directly for the modeling of the data located in Region 2. Region 2 and 3 are named as modeling and testing, respectively. In the modeling region, the GDP per capita is tried to be modeled and the unknown coefficients are found. Note that, the approach uses the previous $l$ values $(P_{i-1}, P_{i-2}, \dots, P_{i-l}$ and corresponding $f(i-1), f(i-2), \dots f(i-l))$ for arbitrary $P_i$ located in Region 2. The third region consists of the data used to test for upcoming predictions. Finally, Region 4 is called the "prediction region" where the aim is to find the GDP per capita values for the time that the actual values have not known yet and implement prediction. The region division is required because there are parameters given in the previous section (Equation (10)) such as $M$, $l$, $\gamma$ which need to be found before the prediction. In Region 2, the modeling is done to find the optimum values of coefficients $a_{kn}$, $M$, $l$, $\gamma$ in Equation (10) for modeling. To model the data, Least Squares Method is employed, which is explained later in this section. After that, one of the purposes of the study is achieved. This is the modeling of the data using the fractional approach. Then, the second purpose comes which is to predict the values of GDP per capita for the upcoming unknown years. In order to find optimum $M$, $l$, $\gamma$ values for the prediction, Region 3, namely testing is needed. In the region, there is an iterative solution where the real discrete data is again known. For instance, in Region 3, it is required to find $f(m_1 + 1)$. Then, by using the proposed method employing the fractional calculus and Least Squares Method, $f(m_1 + 1)$ is obtained with a minimum error by optimizing $M$, $l$, $\gamma$ values for $f(m_1 + 1)$ itself. Then, $f(m_1 + 1)$ is included the dataset for the next test which is done for $f(m_1 + 2)$. This continues up to $f(m)$. Then, with optimized $M$, $l$, $\gamma$, the predicted $f(m_x)$ is found in Region 4.

To model the known data, $f(x)$ representing the data optimally should be obtained. In other words, the unknowns $a_{kn}$, $b_{kn}$ and $f(0)$ in Equation (10) or Equation (11) should be determined. For this, the Least Squares Method is employed.

$$f(i, \gamma) = f(0) + \sum_{k=1}^{l} \sum_{n=1}^{M} a_{kn} C_{kn}(i, \gamma) + \sum_{k=1}^{l} \sum_{n=1}^{M} b_{kn} D_{kn}(i, \gamma). \tag{11}$$

In Equation (12), the squares of total error $\epsilon_T{}^2$ is given. The main purpose of the modeling region is to minimize $\epsilon_T{}^2$ by a gradient-based approach which requires minimization of the square of the total error as the following.

$$\epsilon_T{}^2 = \sum_{i=l}^{m_1} (P_i - f(i, \gamma))^2 \tag{12}$$

$$\frac{\partial \epsilon_T{}^2}{\partial f(0)} = 0,$$

$$\frac{\partial \epsilon_T{}^2}{\partial a_{rt}} = 0,$$

and

$$\frac{\partial \epsilon_T{}^2}{\partial b_{rt}} = 0.$$

where, $r = 1, 2, 3, \ldots l$ and $t = 1, 2, 3, \ldots M$.

It is better to give an example of how to obtain $\frac{\partial \epsilon_T{}^2}{\partial a_{rt}} = 0$ and $\frac{\partial \epsilon_T{}^2}{\partial b_{rt}} = 0$.

$$\frac{\partial \epsilon_T{}^2}{\partial a_{rt}} = 0 \rightarrow \frac{\partial}{\partial a_{rt}} \sum_{i=l}^{m_1} (P_i - f(i, \gamma))^2 = 0$$

Then,

$$2 \sum_{i=l}^{m_1} [P_i - f(i, \gamma)] C_{rt}(i, \gamma) = 0$$

$$\sum_{i=l}^{m_1} C_{rt}(i, \gamma) P_i = f(0) \sum_{i=l}^{m_1} C_{rt}(i, \gamma) + \sum_{i=l}^{m_1} \left\{ \sum_{k=1}^{l} \sum_{n=1}^{M} a_{kn} C_{kn}(i, \gamma) \right\} C_{rt}(i, \gamma)$$

The same procedure is followed for $\frac{\partial \epsilon_T{}^2}{\partial b_{rt}} = 0$.

$$\frac{\partial \epsilon_T{}^2}{\partial b_{rt}} = 0 \rightarrow \frac{\partial}{\partial b_{rt}} \sum_{i=l}^{m_1} (P_i - f(i, \gamma))^2 = 0$$

Then,

$$\sum_{i=l}^{m_1} [P_i - f(i, \gamma)] D_{rt}(i, \gamma) = 0$$

$$\sum_{i=l}^{m_1} D_{rt}(i, \gamma) P_i = f(0) \sum_{i=l}^{m_1} D_{rt}(i, \gamma) + \sum_{i=l}^{m_1} \left\{ \sum_{k=1}^{l} \sum_{n=1}^{M} a_{kn} D_{kn}(i, \gamma) \right\} D_{rt}(i, \gamma)$$

This leads to having a system of linear algebraic equations (SLAE) as given in (13).

$$[A].[B] = [C] \tag{13}$$

where, $[A]$, $[B]$ and $[C]$ is shown in Equations (14), (15) and (16), respectively.

To find $f(x)$ the continuous curve modeling with a minimum error, the optimum fractional-order $\gamma$ is inquired between (0, 1). Then, with optimum fractional-order $\gamma$, the unknown coefficients are determined. In the study, the GDP per capita of Brazil, China, the European Union, India, Italy, Japan, the UK, the USA, Spain and Turkey were used from 1960 until 2018 [38]. The dataset is shown in Tables A1 and A2.

Among them, the year 2018 is in Region 3 as testing to predict for the next years.

Here,

$t$ (years): [1960, 1961, ..., 2018]

$i$ (points): [1, 2, ..., 59]

$P_i$ (value of $i$): $[P_1, P_2, \ldots, P_{59}]$

$P_i$: It shows the actual GDP per capita of each country in each $i^{th}$ year. For example, $P_2$ is the GDP per capita of the country in 1961.

$i$: It stands for the number for each year. For example, $i = 1$ for 1960, $i = 3$ for 1962 and $i = 59$ for 2018.

$$A = \begin{bmatrix} A_{1,1} & A_{1,2} \\ A_{2,1} & A_{2,2} \end{bmatrix} \tag{14}$$

A matrix consists of the matrix set below, where,

$C_{kn}(x,\gamma) = C_{kn}$ and $D_{kn}(x,\gamma) = D_{kn}$

$$A_{1,1} = \begin{bmatrix} m_1 - l + 1 & \sum_{i=l}^{m_1} C_{11} & \cdots & \sum_{i=l}^{m_1} C_{1M} & \sum_{i=l}^{m_1} C_{21} & \cdots & \sum_{i=l}^{m_1} C_{2M} & \cdots & \sum_{i=l}^{m_1} C_{l1} & \cdots & \sum_{i=l}^{m_1} C_{lM} \\ \sum C_{11} & \sum_{i=l}^{m_1} C_{11}C_{11} & \cdots & \sum_{i=l}^{m_1} C_{1M}C_{11} & \sum_{i=l}^{m_1} C_{21}C_{11} & \cdots & \sum_{i=l}^{m_1} C_{2M}C_{11} & \cdots & \sum_{i=l}^{m_1} C_{l1}C_{11} & \cdots & \sum_{i=l}^{m_1} C_{lM}C_{11} \\ \sum C_{12} & \sum_{i=l}^{m_1} C_{11}C_{12} & \cdots & \sum_{i=l}^{m_1} C_{1M}C_{12} & \sum_{i=l}^{m_1} C_{21}C_{12} & \cdots & \sum_{i=l}^{m_1} C_{2M}C_{12} & \cdots & \sum_{i=l}^{m_1} C_{11}C_{12} & \cdots & \sum_{i=l}^{m_1} C_{lM}C_{12} \\ \vdots & \vdots & \vdots & \vdots & \vdots & \vdots & \vdots & \vdots & \vdots & \vdots & \vdots \\ \sum C_{lm} & \sum_{i=l}^{m_1} C_{11}C_{lM} & \cdots & \sum_{i=l}^{m_1} C_{1M}C_{lM} & \sum_{i=l}^{m_1} C_{21}C_{lM} & \cdots & \sum_{i=l}^{m_1} C_{2M}C_{lM} & \cdots & \sum_{i=l}^{m_1} C_{l1}C_{lM} & \cdots & \sum_{i=l}^{m_1} C_{lM}C_{lM} \end{bmatrix}$$

$$A_{2,1} = \begin{bmatrix} \sum D_{11} & \sum_{i=l}^{m_1} C_{11}D_{11} & \cdots & \sum_{i=l}^{m_1} C_{1M}D_{11} & \sum_{i=l}^{m_1} C_{21}D_{11} & \cdots & \sum_{i=l}^{m_1} C_{2M}D_{11} & \cdots & \sum_{i=l}^{m_1} C_{l1}D_{11} & \cdots & \sum_{i=l}^{m_1} C_{lM}D_{11} \\ \sum_{i=l}^{m_1} D_{12} & \sum_{i=l}^{m_1} C_{11}D_{12} & \cdots & \sum_{i=l}^{m_1} C_{1M}D_{12} & \sum_{i=l}^{m_1} C_{21}D_{12} & \cdots & \sum_{i=l}^{m_1} C_{2M}D_{12} & \cdots & \sum_{i=l}^{m_1} C_{l1}D_{12} & \cdots & \sum_{i=l}^{m_1} C_{lM}D_{12} \\ \vdots & \vdots & \vdots & \vdots & \vdots & \vdots & \vdots & \vdots & \vdots & \vdots & \vdots \\ \sum_{i=l}^{m_1} D_{lM} & \sum_{i=l}^{m_1} C_{11}D_{lM} & \cdots & \sum_{i=l}^{m_1} C_{1M}D_{lM} & \sum_{i=l}^{m_1} C_{21}D_{lM} & \cdots & \sum_{i=l}^{m_1} C_{2M}D_{1M} & \cdots & \sum_{i=l}^{m_1} C_{l1}D_{lM} & \vdots & \sum_{i=l}^{m_1} C_{lM}D_{lM} \end{bmatrix}$$

$$A_{1,2} = \begin{bmatrix} \sum_{i=l}^{m_1} D_{11} & \cdots & \sum_{i=l}^{m_1} D_{1M} & \cdots & \sum_{i=l}^{m_1} D_{lM} \\ \sum_{i=l}^{m_1} D_{11}C_{11} & \cdots & \sum_{i=l}^{m_1} D_{1M}C_{11} & \cdots & \sum_{i=l}^{m_1} D_{lM}C_{11} \\ \sum_{i=l}^{m_1} D_{11}C_{12} & \cdots & \sum_{i=l}^{m_1} D_{1M}C_{12} & \cdots & \sum_{i=l}^{m_1} D_{lM}C_{12} \\ \vdots & \vdots & \vdots & \vdots & \vdots \\ \sum_{i=l}^{m_1} D_{11}C_{lM} & \cdots & \sum_{i=l}^{m_1} D_{1M}C_{lM} & \cdots & \sum_{i=l}^{m_1} D_{lM}C_{lM} \end{bmatrix}$$

$$A_{2,2} = \begin{bmatrix} \sum_{i=l}^{m_1} D_{11}D_{11} & \cdots & \sum_{i=l}^{m_1} D_{1M}D_{11} & \cdots & \sum_{i=l}^{m_1} D_{lM}D_{11} \\ \sum_{i=l}^{m_1} D_{11}D_{12} & \cdots & \sum_{i=l}^{m_1} D_{1M}D_{12} & \cdots & \sum_{i=l}^{m_1} D_{lM}D_{12} \\ \vdots & \vdots & \vdots & \vdots & \vdots \\ \sum_{i=l}^{m_1} D_{11}D_{lM} & \cdots & \sum_{i=l}^{m_1} D_{1M}D_{lM} & \cdots & \sum_{i=l}^{m_1} D_{lM}D_{lM} \end{bmatrix}$$

$$[B] = \begin{bmatrix} f(0) & a_{11} & a_{12} & \cdots & a_{1M} & a_{21} & a_{22} & \cdots & a_{2M} & \ldots a_{l1} \ldots & a_{lM} & b_{11} & b_{12} & \cdots & b_{1M} & b_{21} & \ldots & b_{2M} & \cdots & b_{l1} & b_{l2} & \cdots & b_{lM} \end{bmatrix}^T \tag{15}$$

$$[C] = \left[ \; \sum_{i=l}^{m_1} P_i \quad \sum_{i=l}^{m_1} P_i C_{11} \quad \sum_{i=l}^{m_1} P_i C_{12} \quad \ldots \quad \sum_{i=l}^{m_1} P_i C_{lM} \quad \sum_{i=l}^{m_1} P_i D_{11} \quad \sum_{i=l}^{m_1} P_i D_{12} \quad \ldots \quad \sum_{i=l}^{m_1} P_i D_{lM} \; \right]^{T} \quad (16)$$

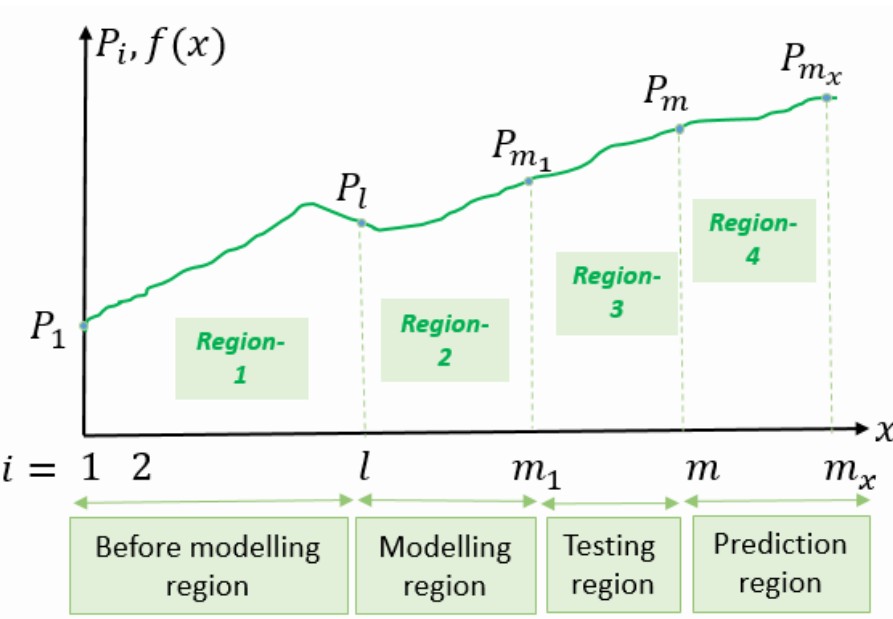

**Figure 1.** The regions of the dataset.

### 3.2. Prediction with Deep Assessment

To find the optimized values of the unknowns for the prediction, the testing region (3rd region) is required. The predictions obtained in the test region $(m_1 < i < m)$ are also given in Table 1. For testing, the data up to $m_1 = 58$ have been taken into consideration in the operations. The $f(m_1 + 1)$ value was found from the obtained modeling. Then, the value is kept, and the next step was started again for $(f(m_1 + 2))$. These operations are done until the last value of the test zone. In our case, $m = 59$.

**Table 1.** Comparison of modeling results ($\gamma$, $M$ and MAPE values) of countries for $l = 10$.

| Country | $\gamma$ Deep Assessment | $\gamma$ Fractional Model-1 | Deep Assessment * ($l<i<m$) | Polynomial Model * ($l<i<m$) | Fractional Model-1 * ($l<i<m$) | $M$ |
|---|---|---|---|---|---|---|
| US | 0.44 | 0.54 | 0.81% | 1.01% | 1.06% | 15 |
| UK | 0.14 | 0.85 | 5.38% | 7.03% | 6.61% | 15 |
| Brazil | 0.06 | 0.58 | 7.26% | 7.13% | 9.00% | 17 |
| China | 0.03 | 0.95 | 2.84% | 5.62% | 5.67% | 11 |
| India | 0.15 | 0.02 | 3.09% | 2.51% | 4.10% | 16 |
| Japan | 0.26 | 0.69 | 4.45% | 4.64% | 5.82% | 20 |
| EU | 0.06 | 0.89 | 4.02% | 3.41% | 5.71% | 20 |
| Italy | 0.39 | 1 | 4.70% | 8.81% | 8.81% | 9 |
| Spain | 0.22 | 0.58 | 4.44% | 6.49% | 6.36% | 13 |
| Turkey | 0.71 | 0.01 | 6.09% | 11.81% | 8.93% | 10 |

*\* $MAPE_{Modeling}$ values.*

The last region is called the "Prediction Region." Here, using Region 1, 2 and 3, the prediction for the upcoming years is obtained. After having modeled and tested regions, the unknowns in Equation (11) have already found in an optimal manner. After testing, Region 4 is started. In the region, the first prediction $f(m + 1)$ is found by using the coefficients and unknowns found by the

testing region. After that, the first predicted value ($f(m+1)$) is included in Region 3 (testing) for the consecutive prediction $f(m+2)$. This procedure is reiterated and recycled up to $f(m_x)$.

The prediction results for 2019 are given in Table 2. For example, as of the end of 2019 ($f(m+2)$), Brazil, China, European Union, India, Italy, Japan, the UK, the USA, Spain and Turkey's GDP per capita values are expected as listed.

**Table 2.** Test ($m_1 < i < m$) results ($\gamma$, $l$, $M$ and MAPE) of GDP per capita for corresponding countries.

| Country | $\gamma$ | $l$ | $M$ | $\gamma$ Interpolation | Deep Assessment * | Deep Learning * |
|---------|----------|-----|-----|------------------------|-------------------|-----------------|
| Brazil | 0.18 | 24 | 3 | 0.32 | 0.1303% | 0.4728% |
| China | 0.97 | 11 | 3 | 0.5 | 0.7147% | 1.6365% |
| India | 0.96 | 3 | 2 | 0.99 | 0.3379%5 | 0.7203% |
| Italy | 0.43 | 20 | 4 | 0.43 | 0.1048% | 3.0796% |
| Japan | 0.57 | 4 | 3 | 1 | 0.3499% | 1.1091% |
| Spain | 0.99 | 2 | 3 | 0.99 | 0.0560% | 1.5683% |
| Turkey | 0.39 | 17 | 4 | 0.39 | 0.1167% | 2.3691% |
| EU | 0.32 | 20 | 5 | 0.22 | 0.1044% | 0.2522% |
| US | 0.39 | 25 | 2 | 0.18 | 0.1081% | 0.8424% |
| UK | 0.18 | 18 | 7 | 0.05 | 0.9129% | 3.0508% |

\* $MAPE_{Prediction}$ values.

In Figure 2, the algorithm for prediction with DAM is illustrated. The first step of the algorithm is to initialize the parameters ($l$, $M$, $x_1$, $x_2$, $\ldots x_m$ and $P_1$, $P_2$, $\ldots P_m$). Then, the counter variable $N$ is introduced, which counts the number of prediction steps. The total number of required predicted steps is denoted as $n_0$. As an initial value, the fractional-order $\gamma$ is assigned 0 and the increment is 0.01 for each loop to find the optimized value. For each value of $\gamma$ between 0 and 1, matrix A given as Equation (14) is created and then, the unknown coefficients given in Equation (10) are calculated. After that, using the actual data in Region 1 and Region 2, the modeling of data between $P_l$ and $P_m$ is actualized for Region 2. Then, the error defined in Equation (12) is calculated. The value of the error is analyzed and compared to previously obtained values. If it is smaller than the previous one, the corresponding fractional-order value is memorized. At the end of Loop II, the optimal value of the fractional-order, which coincides with the optimal modeling is found and corresponding coefficients given in Equation (10) is determined. Then, the prediction for the next forthcoming value is made with Equation (10). After that, all the procedures starting from the increment of $N$ is repeated so that the previously predicted value is added to the initial data for the next step prediction. This process is repeated up to the termination of Loop I. Finally, $n_0$ the number of predictions is obtained. Keep in mind that, for the parameters $l$ and $M$, there exist two loops starting from 1 to $L_0$ and 1 to $M_0$ searching the optimum values of the parameters in order to get the outcomes with a minimum error for the testing region, respectively. Here, $L_0$ and $M_0$ are pre-defined some constant values.

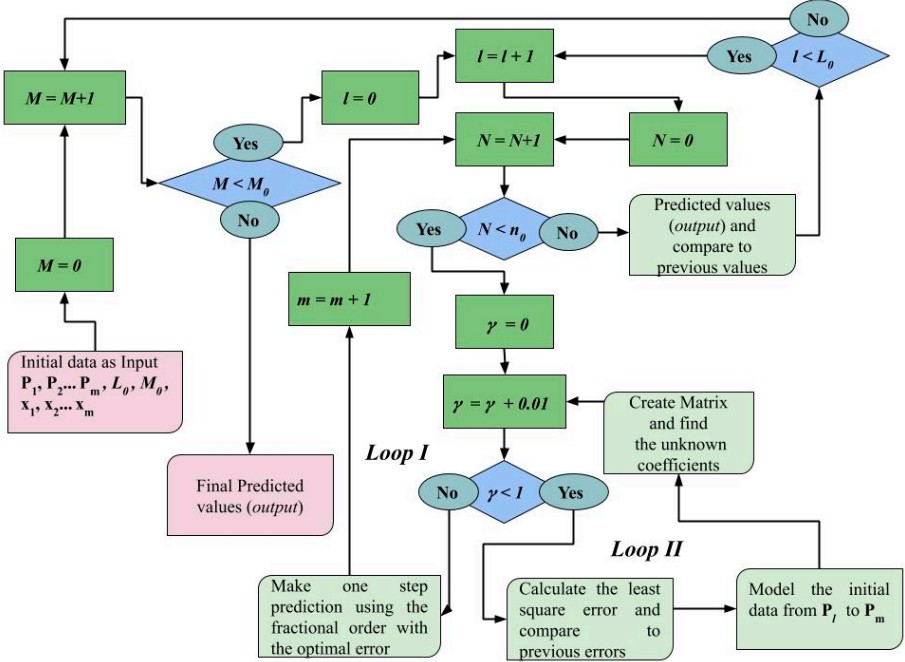

**Figure 2.** The algorithm for the prediction.

### 3.3. Long Short-Term Memory

In our study, we compare the modeling with the polynomial curve fitting method and in the prediction, we compare Deep Assessment with the LSTM method. Conventional neural networks are insufficient for modeling the content of temporal data. Recursive neural networks (RNN) model the sequential structure of data by feeding itself with the output of the previous time step. LSTMs are special types of RNNs that operate over sequences and are used in time series analysis [39]. An LSTM cell has four gates: input, forget, output and gate. With these gates, LSTMs optionally inherit the information from the previous time steps. Forget gate ($f$), input gate ($i$) and output gate ($o$) are sigmoid functions ($\sigma$) and they take values between 0 and 1. Gate $g$ has hyperbolic tangent (*tanh*) activation and is between $-1$ and 1. The Gate and forward propagation equations are listed below as Equations (17)–(22). Here $c_t^l$ and $h_t^l$ refer to cell state and hidden state of layer $l$ at time step $t$, respectively. Each gate takes input from the previous time step ($h_{t-1}^l$) and previous layer ($h_t^{l-1}$) and has its own set of learnable parameters $W$'s and $b$'s.

$$f_t = \sigma\left(W_f[h_{t-1}^l, h_t^{l-1}] + b_f\right) \tag{17}$$

$$i_t = \sigma\left(W_i[h_{t-1}^l, h_t^{l-1}] + b_i\right) \tag{18}$$

$$o_t = \sigma\left(W_o[h_{t-1}^l, h_t^{l-1}] + b_o\right) \tag{19}$$

$$g_t = tanh\left(W_g[h_{t-1}^l, h_t^{l-1}] + b_g\right) \tag{20}$$

$$c_t^l = f \odot c_{t-1}^l + i \odot g \tag{21}$$

$$h_t^l = o \odot \tanh\left(c_t^l\right) \tag{22}$$

Here, $\odot$ is the Hadamard product. Each LSTM neuron in a network may consist of one or more cells. In every time step, every cell updates its own cell state, $c_t^l$. Equation (22) describes how these cells get updated with forget gate and input gate; $f$ gate decides how much of previous cell state that cell should remember while $i$ gate decides how much it should consider the new input from the previous layer. Then, LSTM neuron updates its internal hidden state by multiplying output and squashed version of

$c_t^l$. An LSTM neuron gives outputs only in hidden state information to another LSTM neuron. Gate $o$ and $c_t$ are used internally in the computation of forward time steps [40]. To forecast time series and compare our proposed approach to neural networks, we employed a stacked LSTM model with 2 layers of LSTMs (each having 50 hidden units) and a linear prediction layer. LSTM model is trained with the Adam optimizer [40].

## 4. Numerical Results

In this section, we report the modeling and prediction performance of the Deep Assessment methodology. Further, we compare the proposed method to other modeling and prediction approaches such as Polynomial Model, Fractional Model-1 [34,35] and LSTM. In this section, results are reported with the Mean Average Precision Error (MAPE) metric and calculated as follows:

$$MAPE = \frac{1}{k} \sum_{i=1}^{k} \left| \frac{v(i) - \tilde{v}(i)}{v(i)} \right| \times 100, \qquad (23)$$

where k is the total number of samples, $v(i)$ is the actual value and $\tilde{v}(i)$ is the predicted value for $i^{th}$ sample.

Before presenting the results, it is important to highlight that for modeling, $M_0$ and $l_0$ are taken 20 and 10, respectively whereas for prediction, $M_0$ and $l_0$ are taken 8 and 25, respectively. The number of prediction, $n_0$ is equal to 1.

### 4.1. Modeling Results

In this part, we compare the modeling performance with Polynomial, Fractional Model-1 and Deep Assessment models.

To achieve modeling, $l$ value needs to be investigated. For the modeling of the GDP per capita of each country, the required previous data $l$ of past years used in the algorithm differs after optimization. In order to make a fair evaluation, $l$ value is fixed among all countries to 10. Modeling results for Deep Assessment, Polynomial Model and Fractional Model-1 are shown in Table 1. Optimized $M$ values after processing can be seen in the last column. The Deep Assessment model has a %4.308 average MAPE and outperforms Polynomial and Fractional Model-1 by %1.538 and %1.899 average error rates. All three methods model the US best with %0.81, %1.01 and %1.06 error. Further, in the case of Italy, Fractional Model-1 uses the fractional-order value of 1 and produces %8.81 MAPE, equal to the Polynomial method as expected because for the fractional-order value of 1 is the same with the Polynomial Method. However, DAM yields fractional order of 0.39, decreasing the error to 4.70%, justifying the advantage of employing fractional calculus and previous values of the data itself.

$$MAPE_{Modeling} = \frac{1}{m - l + 1} \sum_{i=l}^{m} \left| \frac{P(i) - f(i, \gamma)}{P(i)} \right| \times 100. \qquad (24)$$

GDP per capita data, Deep Assessment, Polynomial and Fractional Model-1 modeling results are shown in Figure 1 for each country. One can conclude that when data points have high variance all models produce high error rates, as in Turkey and Italy. For Japan and Brazil, DAM (Deep Assessment Method) and Polynomial models produce similar results. Also, it can be seen from the Figure 3, both Deep Assessment and Fractional Model-1 have a low bias when compared to the Polynomial model and overfits to dataset less. This is possible because of the memory property of the proposed approach. Except for Brazil, India and the EU, the proposed method yields superior results compared to other models.

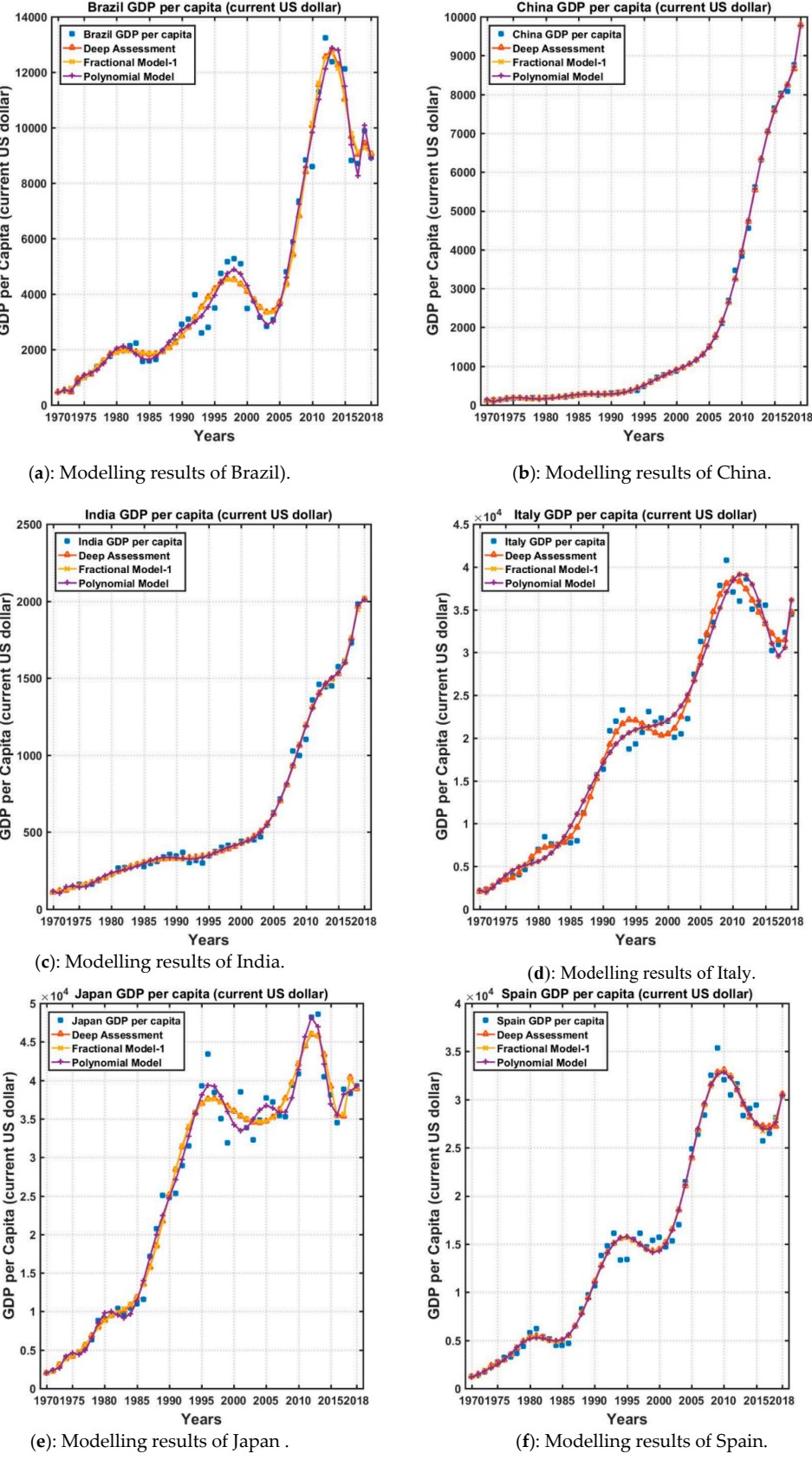

(**a**): Modelling results of Brazil).

(**b**): Modelling results of China.

(**c**): Modelling results of India.

(**d**): Modelling results of Italy.

(**e**): Modelling results of Japan .

(**f**): Modelling results of Spain.

**Figure 3.** *Cont.*

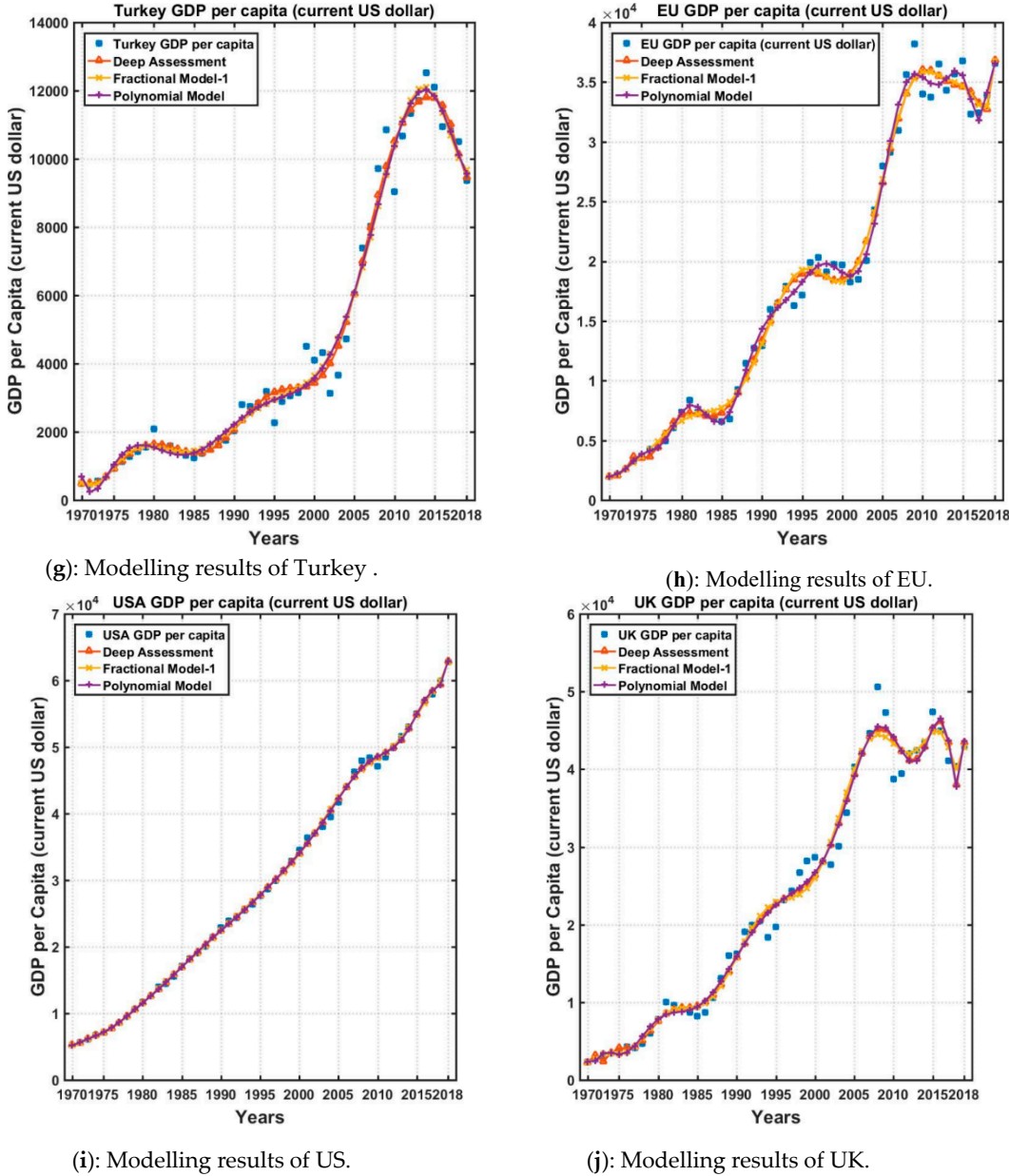

(**g**): Modelling results of Turkey .

(**h**): Modelling results of EU.

(**i**): Modelling results of US.

(**j**): Modelling results of UK.

**Figure 3.** Modelling results of the countries (Brazil, China, India, Italy, Japan, the UK, the USA, Spain and Turkey) and the European Union or Deep Assessment (Blue), Fractional Model 1 (Yellow), Polynomial Model (Purple).

*4.2. Prediction Results*

In this section, we compare the accuracy rate of the prediction of Deep Assessment and Deep Learning models. As in modeling, the GDP per capita dataset is used to assess the performance of the proposed method. Table 2 illustrates optimized $\gamma$, $l$, $M$ values and the corresponding performance of DAM and LSTM. Here, column 6 reports the performance of DAM while column 7 represents LSTM. Column 5 shows that the Deep Assessment methodology predicts GDP per capita with an average 0.29% error with predicting all countries with 1.< (less than 1 percent) of error. The best-predicted country is Spain while UK's prediction is the least accurate with 0.91% error. On the other hand, LSTM yields 1.51% error on average. For both DAM and LSTM, UK yields the highest error. Table 2

demonstrates that in the implemented setting, DAM outperforms LSTM by 1.21% average error and produces fair results.

$$MAPE_{Prediction} = \frac{1}{m_1 - l + 1} \sum_{i=m_1}^{m} \left| \frac{P(i) - f(i,\gamma)}{P(i)} \right| \times 100. \tag{25}$$

Table 3 reports the prediction of GDP per capita for the year 2019 is illustrated in Table 2 for both DAM and LSTM methods. For countries Brazil, China, India, Turkey, the UK and the US, predictions obtained by the two models are similar. On the other hand, Italy and Spain yield different results.

**Table 3.** GDP per Capita Prediction of Countries for 2019 (US dollars).

| Country | Deep Assessment | Deep Learning |
|---------|-----------------|---------------|
| Brazil | 7932 | 8013 |
| China | 10,312 | 10,273 |
| India | 2154 | 1967 |
| Italy | 39,028 | 35,141 |
| Japan | 34,421 | 37,994 |
| Spain | 30,385 | 35,372 |
| Turkey | 8260 | 8920 |
| US | 65,767 | 63,844 |
| UK | 44,897 | 44,702 |
| EU | 40,487 | 36,487 |

## 5. Conclusions

In this study, a model called "Deep Assessment" is introduced which employs Fractional Calculus to model discrete data as the summation of previous values and derivatives. Different to the literature and our previous work, the proposed approach also predicts the incoming values of the discrete data in addition to modeling. The method is evaluated on modeling and predicting GDP per capita, using a dataset including the period of 1960–2018 for nine countries (Brazil, China, European Union, India, Italy, Japan, UK, the USA, Spain and Turkey) and the European Union. Using the fractional differential equation and the summation of previous values for the modeling of GDP per capita at a specific time instant bring non-locality, memory and generalization of the problem for different fractional order. In experiments, first, GDP per capita is modeled. The Deep Assessment model has a 4.308% average MAPE and outperforms Polynomial and Fractional Model-1 by 1.538% and 1.899% average error rates for modeling. For prediction, LSTM, a special type of neural network is used to assess the performance of the model. In the selected test region, it is shown that Deep Assessment is superior to LSTM by 1.51% average error. Results illustrate that the proposed method yields promising results and demonstrates the benefits of combining fractional calculus and differential equations. Evaluation of multivariable and multifunctional problems, analyzing time windows, randomness, noise and error changes are left to future work.

**Author Contributions:** The contribution of each author is listed as follows. E.K. has contributed to supervision, conceptualization, investigation, methodology, and administration. V.T. plays an important role in resources, supervision, and validation. K.K. supported conceptualization, writing, and editing. N.Ö.Ö. was the key person about visualization, investigation, administration, validation, and writing. E.E. has contributed to validation, visualization, writing, and editing. All authors have read and agreed to the published version of the manuscript.

**Funding:** This research was funded by Istanbul Technical University (ITU) Vodafone Future Lab with the grant number ITUVF20180901P11.

**Conflicts of Interest:** The authors declare no conflict of interest. The funders had no role in the design of the study; in the collection, analyses or interpretation of data; in the writing of the manuscript or in the decision to publish the results.

# Appendix A

**Table A1.** GDP per capita (US dollars) values of the countries.

| i | Years | Brazil | China | EU | India | Italy |
|---|---|---|---|---|---|---|
| 1 | 1960 | 210.1099 | 89.52054 | 890.4056 | 82.1886 | 804.4926 |
| 2 | 1961 | 205.0408 | 75.80584 | 959.71 | 85.3543 | 887.3367 |
| 3 | 1962 | 260.4257 | 70.90941 | 1037.326 | 89.88176 | 990.2602 |
| 4 | 1963 | 292.2521 | 74.31364 | 1135.194 | 101.1264 | 1126.019 |
| 5 | 1964 | 261.6666 | 85.49856 | 1245.499 | 115.5375 | 1222.545 |
| 6 | 1965 | 261.3544 | 98.48678 | 1346.058 | 119.3189 | 1304.454 |
| 7 | 1966 | 315.7972 | 104.3246 | 1448.551 | 89.99731 | 1402.442 |
| 8 | 1967 | 347.4931 | 96.58953 | 1546.804 | 96.33914 | 1533.693 |
| 9 | 1968 | 374.7868 | 91.47272 | 1602.06 | 99.87596 | 1651.939 |
| 10 | 1969 | 403.8843 | 100.1299 | 1762.472 | 107.6223 | 1813.388 |
| 11 | 1970 | 445.0231 | 113.163 | 1950.732 | 112.4345 | 2106.864 |
| 12 | 1971 | 504.7495 | 118.6546 | 2195.145 | 118.6032 | 2305.61 |
| 13 | 1972 | 586.2144 | 131.8836 | 2611.729 | 122.9819 | 2671.137 |
| 14 | 1973 | 775.2733 | 157.0904 | 3296.935 | 143.7787 | 3205.252 |
| 15 | 1974 | 1004.105 | 160.1401 | 3685.596 | 163.4781 | 3621.146 |
| 16 | 1975 | 1153.831 | 178.3418 | 4274.046 | 158.0362 | 4106.994 |
| 17 | 1976 | 1390.625 | 165.4055 | 4406.238 | 161.0921 | 4033.099 |
| 18 | 1977 | 1567.006 | 185.4228 | 4968.988 | 186.2135 | 4603.6 |
| 19 | 1978 | 1744.257 | 156.3964 | 6064.883 | 205.6934 | 5610.498 |
| 20 | 1979 | 1908.488 | 183.9832 | 7377.165 | 224.001 | 6990.286 |
| 21 | 1980 | 1947.276 | 194.8047 | 8384.718 | 266.5778 | 8456.919 |
| 22 | 1981 | 2132.883 | 197.0715 | 7391.077 | 270.4706 | 7622.833 |
| 23 | 1982 | 2226.767 | 203.3349 | 7093.702 | 274.1113 | 7556.523 |
| 24 | 1983 | 1570.54 | 225.4319 | 6859.966 | 291.2381 | 7832.575 |
| 25 | 1984 | 1578.926 | 250.714 | 6572.019 | 276.668 | 7739.715 |
| 26 | 1985 | 1648.082 | 294.4588 | 6775.647 | 296.4352 | 7990.687 |
| 27 | 1986 | 1941.491 | 281.9281 | 9265.924 | 310.4659 | 11,315.02 |
| 28 | 1987 | 2087.308 | 251.812 | 11,432.23 | 340.4168 | 14,234.73 |
| 29 | 1988 | 2300.377 | 283.5377 | 12,711.96 | 354.1493 | 15,744.66 |
| 30 | 1989 | 2908.496 | 310.8819 | 12,936.46 | 346.1129 | 16,386.66 |
| 31 | 1990 | 3100.28 | 317.8847 | 15,989.22 | 367.5566 | 20,825.78 |
| 32 | 1991 | 3975.39 | 333.1421 | 16,496.51 | 303.0556 | 21,956.53 |
| 33 | 1992 | 2596.92 | 366.4607 | 17,919.02 | 316.9539 | 23,243.47 |
| 34 | 1993 | 2791.209 | 377.3898 | 16,256.42 | 301.159 | 18,738.76 |
| 35 | 1994 | 3500.611 | 473.4923 | 17,194.12 | 346.103 | 19,337.63 |
| 36 | 1995 | 4748.216 | 609.6567 | 19,898.44 | 373.7665 | 20,664.55 |
| 37 | 1996 | 5166.164 | 709.4138 | 20,295.17 | 399.9501 | 23,081.6 |
| 38 | 1997 | 5282.009 | 781.7442 | 19,121.21 | 415.4938 | 21,829.35 |
| 39 | 1998 | 5087.152 | 828.5805 | 19,763.51 | 413.2989 | 22,318.14 |
| 40 | 1999 | 3478.373 | 873.2871 | 19,698.89 | 441.9988 | 21,997.62 |
| 41 | 2000 | 3749.753 | 959.3725 | 18,261.97 | 443.3142 | 20,087.59 |
| 42 | 2001 | 3156.799 | 1053.108 | 18,457.89 | 451.573 | 20,483.22 |
| 43 | 2002 | 2829.283 | 1148.508 | 20,055.33 | 470.9868 | 22,270.14 |
| 44 | 2003 | 3070.91 | 1288.643 | 24,310.25 | 546.7266 | 27,465.68 |
| 45 | 2004 | 3637.462 | 1508.668 | 27,960.05 | 627.7742 | 31,259.72 |
| 46 | 2005 | 4790.437 | 1753.418 | 29,115.63 | 714.861 | 32,043.14 |
| 47 | 2006 | 5886.464 | 2099.229 | 30,960.56 | 806.7533 | 33,501.66 |
| 48 | 2007 | 7348.031 | 2693.97 | 35,630.94 | 1028.335 | 37,822.67 |
| 49 | 2008 | 8831.023 | 3468.304 | 38,185.62 | 998.5223 | 40,778.34 |
| 50 | 2009 | 8597.915 | 3832.236 | 34,019.28 | 1101.961 | 37,079.76 |
| 51 | 2010 | 11,286.24 | 4550.454 | 33,740.65 | 1357.564 | 36,000.52 |
| 52 | 2011 | 13,245.61 | 5618.132 | 36,506.64 | 1458.104 | 38,599.06 |
| 53 | 2012 | 12,370.02 | 6316.919 | 34,328.82 | 1443.88 | 35,053.53 |
| 54 | 2013 | 12,300.32 | 7050.646 | 35,683.86 | 1449.606 | 35,549.97 |
| 55 | 2014 | 12,112.59 | 7651.366 | 36,787.23 | 1573.881 | 35,518.42 |
| 56 | 2015 | 8814.001 | 8033.388 | 32,319.45 | 1605.605 | 30,230.23 |
| 57 | 2016 | 8712.887 | 8078.79 | 32,425.13 | 1729.268 | 30,936.13 |
| 58 | 2017 | 9880.947 | 8759.042 | 33,908 | 1981.269 | 32,326.84 |
| 59 | 2018 | 8920.762 | 9770.847 | 36,569.73 | 2009.979 | 34,483.2 |

**Table A2.** GDP per capita (US dollars) values of the countries.

| i | Years | Japan | Spain | UK | US | Turkey |
|---|---|---|---|---|---|---|
| 1 | 1960 | 478.9953 | 396.3923 | 1397.595 | 3007.123 | 509.4239 |
| 2 | 1961 | 563.5868 | 450.0533 | 1472.386 | 3066.563 | 283.8283 |
| 3 | 1962 | 633.6403 | 520.2061 | 1525.776 | 3243.843 | 309.4467 |
| 4 | 1963 | 717.8669 | 609.4874 | 1613.457 | 3374.515 | 350.6629 |
| 5 | 1964 | 835.6573 | 675.2416 | 1748.288 | 3573.941 | 369.5834 |
| 6 | 1965 | 919.7767 | 774.7616 | 1873.568 | 3827.527 | 386.3581 |
| 7 | 1966 | 1058.504 | 889.6599 | 1986.747 | 4146.317 | 444.5494 |
| 8 | 1967 | 1228.909 | 968.3068 | 2058.782 | 4336.427 | 481.6937 |
| 9 | 1968 | 1450.62 | 950.5457 | 1951.759 | 4695.923 | 526.2135 |
| 10 | 1969 | 1669.098 | 1077.679 | 2100.668 | 5032.145 | 571.6178 |
| 11 | 1970 | 2037.56 | 1212.289 | 2347.544 | 5234.297 | 489.9303 |
| 12 | 1971 | 2272.078 | 1362.166 | 2649.802 | 5609.383 | 455.1049 |
| 13 | 1972 | 2967.042 | 1708.809 | 3030.433 | 6094.018 | 558.421 |
| 14 | 1973 | 3997.841 | 2247.553 | 3426.276 | 6726.359 | 686.4899 |
| 15 | 1974 | 4353.824 | 2749.925 | 3665.863 | 7225.691 | 927.7991 |
| 16 | 1975 | 4659.12 | 3209.837 | 4299.746 | 7801.457 | 1136.375 |
| 17 | 1976 | 5197.807 | 3279.313 | 4138.168 | 8592.254 | 1275.956 |
| 18 | 1977 | 6335.788 | 3627.591 | 4681.44 | 9452.577 | 1427.372 |
| 19 | 1978 | 8821.843 | 4356.439 | 5976.938 | 10,564.95 | 1549.644 |
| 20 | 1979 | 9105.136 | 5770.215 | 7804.762 | 11,674.19 | 2079.22 |
| 21 | 1980 | 9465.38 | 6208.578 | 10,032.06 | 12,574.79 | 1564.247 |
| 22 | 1981 | 10,361.32 | 5371.166 | 9599.306 | 13,976.11 | 1579.074 |
| 23 | 1982 | 9578.114 | 5159.709 | 9146.077 | 14,433.79 | 1402.406 |
| 24 | 1983 | 10,425.41 | 4478.5 | 8691.519 | 15,543.89 | 1310.256 |
| 25 | 1984 | 10,984.87 | 4489.989 | 8179.194 | 17,121.23 | 1246.825 |
| 26 | 1985 | 11,584.65 | 4699.656 | 8652.217 | 18,236.83 | 1368.401 |
| 27 | 1986 | 17,111.85 | 6513.503 | 10,611.11 | 19,071.23 | 1510.677 |
| 28 | 1987 | 20,745.25 | 8239.614 | 13,118.59 | 20,038.94 | 1705.895 |
| 29 | 1988 | 25,051.85 | 9703.124 | 15,987.17 | 21,417.01 | 1745.365 |
| 30 | 1989 | 24,813.3 | 10,681.97 | 16,239.28 | 22,857.15 | 2021.859 |
| 31 | 1990 | 25,359.35 | 13,804.88 | 19,095.47 | 23,888.6 | 2794.35 |
| 32 | 1991 | 28,925.04 | 14,811.9 | 19,900.73 | 24,342.26 | 2735.708 |
| 33 | 1992 | 31,464.55 | 16,112.19 | 20,487.17 | 25,418.99 | 2842.37 |
| 34 | 1993 | 35,765.91 | 13,339.91 | 18,389.02 | 26,387.29 | 3180.188 |
| 35 | 1994 | 39,268.57 | 13,415.29 | 19,709.24 | 27,694.85 | 2270.338 |
| 36 | 1995 | 43,440.37 | 15,471.96 | 23,123.18 | 28,690.88 | 2897.866 |
| 37 | 1996 | 38,436.93 | 16,109.08 | 24,332.7 | 29,967.71 | 3053.947 |
| 38 | 1997 | 35,021.72 | 14,730.8 | 26,734.56 | 31,459.14 | 3144.386 |
| 39 | 1998 | 31,902.77 | 15,394.35 | 28,214.27 | 32,853.68 | 4496.497 |
| 40 | 1999 | 36,026.56 | 15,715.33 | 28,669.54 | 34,513.56 | 4108.123 |
| 41 | 2000 | 38,532.04 | 14,713.07 | 28,149.87 | 36,334.91 | 4316.549 |
| 42 | 2001 | 33,846.47 | 15,355.7 | 27,744.51 | 37,133.24 | 3119.566 |
| 43 | 2002 | 32,289.35 | 17,025.53 | 30,056.59 | 38,023.16 | 3659.94 |
| 44 | 2003 | 34,808.39 | 21,463.44 | 34,419.15 | 39,496.49 | 4718.2 |
| 45 | 2004 | 37,688.72 | 24,861.28 | 40,290.31 | 41,712.8 | 6040.608 |
| 46 | 2005 | 37,217.65 | 26,419.3 | 42,030.29 | 44,114.75 | 7384.252 |
| 47 | 2006 | 35,433.99 | 28,365.31 | 44,599.7 | 46,298.73 | 8035.377 |
| 48 | 2007 | 35,275.23 | 32,549.97 | 50,566.83 | 47,975.97 | 9711.874 |
| 49 | 2008 | 39,339.3 | 35,366.26 | 47,287 | 48,382.56 | 10,854.17 |
| 50 | 2009 | 40,855.18 | 32,042.47 | 38,713.14 | 47,099.98 | 9038.52 |
| 51 | 2010 | 44,507.68 | 30,502.72 | 39,435.84 | 48,466.82 | 10,672.39 |
| 52 | 2011 | 48,168 | 31,636.45 | 42,038.5 | 49,883.11 | 11,335.51 |
| 53 | 2012 | 48,603.48 | 28,324.43 | 42,462.71 | 51,603.5 | 11,707.26 |
| 54 | 2013 | 40,454.45 | 29,059.55 | 43,444.56 | 53,106.91 | 12,519.39 |
| 55 | 2014 | 38,109.41 | 29,461.55 | 47,417.64 | 55,032.96 | 12,095.85 |
| 56 | 2015 | 34,524.47 | 25,732.02 | 44,966.1 | 56,803.47 | 10,948.72 |
| 57 | 2016 | 38,794.33 | 26,505.62 | 41,074.17 | 57,904.2 | 10,820.63 |
| 58 | 2017 | 38,331.98 | 28,100.85 | 40,361.42 | 59,927.93 | 10,513.65 |
| 59 | 2018 | 39,289.96 | 30,370.89 | 42,943.9 | 62,794.59 | 9370.176 |

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
