# Peer review of "Deep Assessment Methodology Using Fractional Calculus on Mathematical Modeling and Prediction of Gross Domestic Product per Capita of Countries"

_mathematics, doi:10.3390/math8040633_

Round 1
Reviewer 1 Report
1. "Second, expressing an arbitrary function g(x)
as the summation of polynomials known as Taylor Expansion".
The statement is mathematically incorrect.
In mathematics, it is well known that NOT any function can be represented as a Taylor series
2. Equation (2.5) and (2.6) has misprints.
(d^{gamma}; sin second term n should be n-1).
3. In the general case, the fractional derivative of the Caputo
cannot be represented in the form (2.6).
Equation 2.6 without misprint (i.e. with n-1 in second term) can be used
is only first derivatibe gamma=1.
4. The proposed method and formulas should be described in detail.
Proposed approximations must be substantiated.
The smallness of the remainder term must be proved.
5. Table A1: (GDP per capita (US dollars) values of the countries)
should be also represented as the Plots (in graph form).
6. Line 188.
The presentation of the matrix A should be simplified,
for example, by splitting into blocks and writing as a block matrix.
7. Economic justification of the standard model and the proposed model should be written in great detail.
8. Economic consequences and results in the economics should be described in detail.
Author Response
Dear Sir/Madam, You can find the reply to your precious review in the attachment.
Best regards

Reviewer 2 Report
The paper “Deep Assessment Methodology Using Fractional Calculus on Mathematical Modeling and Prediction of Gross Domestic Product per Capita of Countries ” deals with the usage of the least squares method in combination with the fractional calculus based modelling and prediction and the deep assessment method. The gross domestic product per capita data is modelled as a function satisfying the fractional differential equation.
Besides some comments to the authors I think the paper meets the quality criterion of the journal.
#1
The abstract is not very clear, repeating the text twice. The authors should think of rewriting it, emphasising the new aspect of their research.
#2
The sources mentioned in the introduction could be extended, using newer references.
The sources [12-21] should be extended citing new works by authors such as Richard Magin (biomedical applications), YanQuan Chen (modelling), Ivo Petras (control), Tomas Skovranek (diffusion) etc.
The sources [22-23] should be extended citing works by authors of the MDPI Mathematics, Special Issue - Mathematical Economics: Application of Fractional Calculus
https://www.mdpi.com/journal/mathematics/special_issues/Mathematical_Economics
#3
In conclusion the authors state that: “The study is a first attempt to combine the data modeling and prediction using the fractional calculus and Least Square Method.”
I suggest to emhasise here, that the work given below, was trully the first attemt using LSM in combination with fractional calculus for the economic modelling:
Modeling of the national economies in state-space: A fractional calculus approach
https://doi.org/10.1016/j.econmod.2012.03.019
And the references below are presenting the state-of-art method for fractional calculus based prediction:
One-parameter fractional linear prediction
https://doi.org/10.1016/j.compeleceng.2018.05.020
Optimal fractional linear prediction with restricted memory
http://doi.org/10.1109/LSP.2019.2908278
Thus the authors should reformulate this part and mention the above references.
Author Response
Dear Sir/Madam, You can find a reply to your precious review in the attachment.
Best regards

Reviewer 3 Report
This paper proposes what amounts to a sort of autoregressive model for the time series of the GDP of several countries, without any exogenous variables. Fractional derivatives are used; this can be justified with several arguments, though the paper says only (to sum it up) that fractional models have been used elsewhere and particularly in some economic growth models. What is rather curious here is the absence of exogenous variables, a matter that however deserves no comment at all. Apparently the knowledge of the past evolution of the economy should suffice to predict its future; no white noise or otherwise random input, no other variable intervening. While there are possible arguments (and references) in favour of this approach, none is given. And of course results thus obtained should be compared with those of models with such inputs.
Even the variable modelled seems not to be the one actually desired. The GDP per capita «is a very reasonable and good measurement of a country’s living quality and standard» especially when accounting for purchasing power parity, which does not seem to be the case. Constant prices would also have been a good idea.
Abstract and introduction are rather poorly written, in what concerns both form and content. The list of countries addressed is given twice. The width of the four regions is hard to find.
In short, it is my belief that this paper needs very profound improvements before it can be considered for publication, changes that amount in fact to becoming a different paper.
Author Response

(The authors gave the same response as above.)

Round 2
Reviewer 1 Report
The manuscript can be published.
Reviewer 2 Report
After reading the corrected paper, I suggest to add one more reference, where modelling (and prediction based on the model) of economic data was proposed:
https://doi.org/10.3390/math7070589
Otherwise, the paper was sufficiently improved and can be published after checking and correcting English.
Reviewer 3 Report
The paper has been very significantly improved. I congratulate the authors about this.